# Influence of Nitrogen Sources on D-Lactic Acid Biosynthesis by *Sporolactobacillus laevolacticus* DSM 442 Strain

Alicja Katarzyna Michalczyk [1],*(ID), Sylwia Garbaczewska [1], Bolesław Morytz [1](ID), Arkadiusz Białek [2] and Jerzy Zakrzewski [1],*

[1] Department of Chemical Technology and Biotechnology, Łukasiewicz Research Network-Institute of Industrial Organic Chemistry, Annopol 6, 03-236 Warsaw, Poland; sylwiagarbaczewska@wp.pl (S.G.); boleslaw.morytz@ipo.lukasiewicz.gov.pl (B.M.)

[2] Agrochemical Solutions Hortulanus, Sygietyńskiego 25, 96-316 Stare Budyhort, Poland; hortulanus@hortulanus.agro.pl

* Correspondence: Alicja.Michalczyk@ipo.lukasiewicz.gov.pl (A.K.M.); jerzy_z@wp.pl (J.Z.)

**Abstract:** The purpose of this study was to explore the possibility of replacing an expensive yeast extract contained in the fermentation medium for D-lactic acid (D-LA, R-lactic acid) biosynthesis with an alternative nitrogen source. The screening studies were conducted under stationary conditions and showed that pea seed hydrolysate was the most beneficial substrate in the process of D-LA biosynthesis by the strain *Sporolactobacillus laevolacticus* DSM 442 among the used inorganic and organic nitrogen sources, waste materials, food and agricultural products. After 96 h, 75.5 g/L D-LA was obtained in batch cultures in a medium containing pea seed hydrolysate, with an average productivity of 0.79 g/L/h, yield of 75.5%, and optical purity of 99.4%. In batch cultures fed once, in a medium with an analogous composition, 122.6 g/L D-LA was obtained after 120 h, and the average yield, productivity and optical purity were 87.6%, 1.021 g/L/h, and 99.6%, respectively. Moreover, the amount of D-LA obtained in the fermentation medium enriched with the above-mentioned cheap agricultural product was similar to the amounts obtained in the medium containing yeast extract in both stationary and bioreactor cultures. Our research shows that hydrolyzed pea seeds, which belong to the legume family, may be a promising nitrogen source for the production of D-LA on an industrial scale.

**Keywords:** D-lactic acid; biosynthesis; *Sporolactobacillus laevolacticus* DSM 442; nitrogen source; agricultural products

## 1. Introduction

Lactic acid (LA) is an organic acid produced on an industrial scale. Its production primarily involves the participation of microorganisms, mainly lactic acid bacteria. Due to its properties and wide range of applications (in the food, pharmaceutical, cosmetic, chemical industry), it is a subject of intensive research. Numerous attempts have been made to improve the lactic fermentation process; these have focused mainly on improving the performance and reducing the costs of obtaining LA in biotechnological processes. This includes the selection of appropriate production strains, depending on the desired characteristics of the final fermentation product (L or D-LA isomer), or their improvement, e.g., through genetic engineering, and the use of low-cost, renewable, commonly available raw materials for low-cost processing and the production of LA [1–10]. Reducing the costs of LA production is important in the context of the growing demand for biodegradable plastics, especially polylactide (PLA: polymers synthesized from both D- as well as L-lactic acid), which is known as the material of the 21st century.

PLA is usually produced from optically pure L-lactic acid. However, the utility of this polymer is limited by its low melting temperature (180 °C) [11], thus, poly(D-lactic acid) is also used. Developing a stereocomplex of poly (L-lactic acid) and poly (D-lactic acid) represents an

improvement in the process of PLA synthesis, as the melting point of such polymers rises to over 230 °C. According to Fukushima et al. [12], stereocomplexation improves the mechanical properties, thermal resistance and hydrolytic resistance of PLA-based materials. This opens a huge potential market for D-lactic acid (D-LA, R-lactic acid).

Apart from medical applications, PLA is used in agri-food production to manufacture biodegradable garden films, agrotextiles, pots undergoing decomposition when put in soil with the plant, packaging and disposable items (shopping bags, garbage bags, beverage bottles, packaging for dairy products, cutlery and crockery) and the production of matrices for controlled release of fertilizers and pesticides [13,14]. Therefore, the future of biotechnological production of both stereoisomers of LA is inextricably linked with the use of waste and agricultural products as a source of carbon and nitrogen.

According to the literature, yeast extract is most often used as a nitrogen source in the lactic fermentation process, and it accounts for 38% of the total process costs; therefore, it is a major factor affecting the economy of LA production [15]. Among the various complex nitrogen sources, yeast extract is the best choice for both microbial growth and LA production, although the high cost of yeast extract hinders its use in large-scale industrial procedures [16].

In order to reduce the costs of the process, attempts have been made to use cheap raw materials that are an alternative source of nitrogen, for example, from fish processing [17,18], slaughter by-products [19,20], plant-origin products [15,21–27], agricultural waste [28–31] and by-products of the dairy industry [32,33]. The research conducted so far has mainly focused on the search for cheap yeast extract substitutes for the preparation of L-LA isomer. However, there are few papers available on the possibility of obtaining the D-isomer of LA from these types of products. According to Li et al. [25] and Bai et al. [26], it is possible to use cotton seed hydrolysate or cotton seed flour as a nitrogen component of the fermentation medium for the biosynthesis of D-LA. In turn, Wang et al. [22] pointed to hydrolyzed peanut flour as an effective substitute for yeast extract, and de la Torre et al. [31] mentioned corn steep. In numerous studies, it was not always possible to use the selected component as the sole source of nitrogen in the medium. The obtained amounts of LA on this type of media were often unsatisfactory. Therefore, additional supplementation with a small amount of yeast extract was necessary [16,23,34].

Due to the necessity of further research into the development of efficient and economical process of LA biosynthesis, our study involved searching for production media components that are waste materials from the food industry, cheap and generally available in Poland, or agricultural products that can replace the expensive yeast extract in D-LA biosynthesis process by *Sporolactobacillus laevolacticus* DSM 442 strain. The choice of *Sp. laevolacticus* DSM 442 as a strain enabling the production of D-LA was dictated by its low nutritional requirements during the fermentation process. For more detailed information, see Section 4.

In this study, we found that the strain *Sp. laevolacticus* DSM 442 can efficiently produce D-LA using pea seeds as the sole nitrogen source. Our study may have more general implications for the future strategies of industrial chemical production.

## 2. Materials and Methods

### 2.1. Materials

The following substances were used as the experimental materials: inorganic nitrogen sources, $(NH_4)_2SO_4$ and $NH_4NO_3$ (purchased from POCh, Gliwice, Poland); standard components of culture media, including yeast extract, urea, meat extract, peptone, casein tryptone (purchased from Biocorp, Warszawa, Poland); waste materials, including yeast cake, corn steep liquor, and food; and agricultural products, including baker's yeast, wheat meal, hazelnuts, pumpkin, flax, sunflower, soybean, pea, alfalfa and lupine seeds. All substances except the inorganic salts and standard components of culture media were purchased from a local market. All nitrogen sources were used in quantities equivalent to the nitrogen content in 5 g of yeast extract (0.5 g N). The nitrogen content in the used

waste raw materials, food and agricultural products was determined by the Kjeldahl method [35]. Uncrushed substrates were subjected to multi-stage grinding to the level of approx. 1–2 mm. Commercially available neutral protease (Neutrase EC 3.4.24.28), with an activity of $5 \times 10^4$ U/g according to the manufacturer's data, was purchased from Novozymes A/S (Bagsværd, Denmark). All other chemicals were analytical grade and commercially available.

## 2.2. Microorganism

*Sp. laevolacticus* DSM442 was used as a homofermentative D-LA producer in this study. The strain was purchased from Deutsche Sammlung von Mikroorganismen und Zellkulturen GmbH (DSMZ), Leibniz-Institut DSMZ, Braunschweig, Germany. Stock cultures (1 mL) were stored at $-20$ °C in *Lactobacilli* MRS medium [36] with 25% (*v/v*) glycerol.

## 2.3. Culture Conditions

The inoculum was prepared by transferring glycerol stock culture (1 mL) to an Erlenmeyer flask containing 30 mL of liquid MRS medium for preculture. The flask was subsequently incubated for 48 h at 30 °C without agitation. Then, the culture in the Erlenmeyer flask or bioreactor containing the production medium was inoculated.

## 2.4. Enzymatic Hydrolysis of Waste, Agricultural and Food Raw Materials

The appropriate amount of raw material (Table 1) was suspended in 1 L of distilled water, adjusted to pH = 7, and hydrolyzed using 0.5 g/L of protease. The obtained solution was incubated for approximately 24 h at 45 °C. The produced hydrolysates were used to prepare fermentation media, thus replacing the yeast extract with a new nitrogen source.

**Table 1.** Nitrogen content in selected inorganic and organic raw materials.

| Type of Raw Material | Total Nitrogen Content (%) (*w/w*) | Quantity of Substrate Used in the Production Medium (g/L) |
|---|---|---|
| $(NH_4)_2SO_4$ | 21 [a] | 2.4 |
| $NH_4NO_3$ | 35 [a] | 1.4 |
| Urea | 47 [a] | 1.1 |
| Yeast extract | 10 [b] | 5.0 |
| Meat extract | 10 [b] | 5.0 |
| Peptone Bacteriological | 12 [b] | 4.2 |
| Casein tryptone | 10 [b] | 5.0 |
| Yeast cake | 8.5 [c] | 5.9 |
| Corn steep liquor | 1.9 [c] | 27.0 |
| Baker's yeast | 2.7 [c] | 18.5 |
| Wheat middling | 10 [c] | 5.0 |
| Hazelnuts | 4.1 [c] | 12.3 |
| Pumpkin seeds | 5.5 [c] | 9.0 |
| Flax seeds | 3.2 [c] | 15.6 |
| Sunflower seeds | 2.3 [c] | 21.7 |
| Alfalfa seeds | 9.1 [c] | 5.5 |
| Lupine seeds | 6.5 [c] | 7.8 |
| Pea seeds | 4.1 [c] | 12.3 |
| Soybean seeds | 8.1 [c] | 6.2 |

[a] alculated value, [b] nitrogen content provided by a supplier, [c] nitrogen determination by the Kjeldahl method (*n* = 3).

## 2.5. Influence of Various Nitrogen Sources on D-LA Biosynthesis

Fermentation media containing the organic and inorganic nitrogen source in the amount specified in Table 1 as well as 100 g/L of glucose, 60g/L $CaCO_3$, 2 g/L $MgSO_4 \times 7 H_2O$, 0.1 g/L $MnSO_4 \times 5 H_2O$, 0.1 g/L $FeSO_4 \times 7 \cdot H_2O$ were prepared [1]. Portions of 30 mL of the media were poured into 100 mL Erlenmeyer flasks and sterilized. The flasks

were inoculated with a 48-h culture of *Sp. laevolacticus* DSM 442 strain in the amount of 17% *v/v* and incubated at 37 °C for 7 days under stationary conditions. Two samples were collected from each flask at 24-h intervals to determine the LA (D-LA, L-LA), glucose and by-products content in the post-fermentation broth. The results were expressed in g/L. Each experiment was repeated three times, and the results are the mean value.

### 2.6. Influence of Pea Seeds Hydrolysate and Unhydrolysed Pea Seeds on D-LA Biosynthesis

Fermentation media based on pea seed hydrolysate were prepared in the amount of 12.4 g/L, 24.7 g/L, 37.1 g/L, 49.4 g/L, 61.8 g/L, corresponding to 0.5, 1, 1.5, 2, 2.5 g/L of nitrogen. Analogous media were prepared using pea seeds that had not been subjected to hydrolysis using protease. Portions of 30 mL of the media were poured into 100 mL Erlenmeyer flasks and sterilized. The flasks were inoculated with a 48-h culture of *Sp. laevolacticus* DSM 442 strain in the amount of 17% *v/v* and incubated at 37 °C. After a 5-day fermentation process, two samples were taken from each flask to determine the content of LA (D-LA, L-LA) in the post-fermentation broth. The results were expressed in g/L. Each experiment was repeated three times, and the results are the mean value.

### 2.7. Batch Fermentation

The batch fermentation process was carried out in a BioFlo 320 bioreactor (New Brunswick, Ramsey, MN, USA) of 2 L capacity. The following parameters were maintained: stirring speed, 150 rpm; air flow, 0.8 vvm; and temperature, 37 °C (optimized parameters). The production medium was prepared as follows. Glucose (100 g), $CaCO_3$ (60 g), $MgSO_4 \times 7\,H_2O$ (2 g), $MnSO_4 \times 5\,H_2O$ (0.1 g), $FeSO_4 \times 7{\cdot}H_2O$ (0.1 g) were placed in the reactor. Hydrolyzed pea seeds (prepared according to the procedure described in Section 2.4) were added to bring the final volume to one liter and then sterilized. A 10% (*v/v*) inoculum grown in the MRS medium (100 mL prepared according to the procedure described in Section 2.3) was used to inoculate the production medium in the bioreactor. The fermentation process was carried out until glucose was not detected. pH was maintained at about 5.5. In order to determine the content of LA (D-LA, L-LA), unreacted glucose and possible by-products (acetic acid), HPLC analysis and enzymatic test were performed at 24, 42, 48, 66, 72, 90, 96 h of the fermentation process. Each experiment was repeated three times, and the obtained results are the mean value. The process using the yeast extract was performed in an analogous way.

The maximum content of D-LA in a post-fermentation broth: pea hydrolysate as a nitrogen source, 96 h, 75.5 g (75.5%); yeast extract as a nitrogen source, 72 h, 72.6 g (72.6%).

### 2.8. Fed-Batch Fermentation

The fed-batch fermentation process was carried out in a BioFlo 415 bioreactor (New Brunswick, Ramsey, MN, USA) of 7.5 L capacity. The following parameters were maintained: stirring speed, 150 rpm; air flow, 0.8 vvm; and temperature, 37 °C (optimized parameters). Glucose (300 g), $CaCO_3$ (180 g), $MgSO_4 \times 7\,H_2O$ (6 g), $MnSO_4 \times 5\,H_2O$ (0.3 g), $FeSO_4 \times 7{\cdot}H_2O$ (0.3 g) were placed in the reactor. Hydrolyzed pea seeds prepared according to the procedure described in Section 2.4 was added to bring the final volume to three liters and then sterilized. A 10% (*v/v*) inoculum grown in the MRS medium (300 mL prepared according to the procedure described in Section 2.3) was used to inoculate the production medium in the bioreactor. During the fermentation process, when the glucose concentration decreased below 20 g/L, a fresh, sterilized one liter portion of the medium containing glucose (260 g) and $CaCO_3$ (156 g), $MgSO_4 \times 7\,H_2O$ (2 g), $MnSO_4 \times 5\,H_2O$ (0.1 g), $FeSO_4 \times 7{\cdot}H_2O$ (0.1 g) was added once to increase the glucose concentration in the medium at the level of 70–80 g/L. The pH was maintained at about 5.5. The fermentation process was carried out until glucose was not detected. In order to determine the content of LA (D-LA, L-LA), unreacted glucose and possible by-products (acetic acid), HPLC analysis and enzymatic test were performed at 24, 42, 48, 66, 72, 90, 96, 114, 120, 136 h of the fermen-

tation process. Each experiment was repeated three times, and the obtained results are the mean value. The process using the yeast extract was performed in an analogous way.

The maximum content of D-LA in a post-fermentation broth: pea hydrolysate as a nitrogen source, 120 h, 490.4 g (87.6%); yeast extract as a nitrogen source, 96 h, 546.5 g (97.5%).

### 2.9. Analysis of Biosynthesis Products

Samples were treated with one volume of 0.5 M $H_2SO_4$ and centrifuged at $8000\times g$ (Universal 320R, Hettich, Germany) for 30 min. The supernatants were filtered through a 0.22 μm membrane (Millipore, USA) and used for analysis. The concentrations of LA, and glucose and organic acid (acetic acid) as by-products were determined by high performance liquid chromatography (HPLC, Shimadzu, Japan) equipped with a differential refracting index detector (RID-20A), UV detector (SPD-M20A) and an ion exclusion column (Aminex, HPX 87H, Bio-Rad, Hercules, CA, USA). The analytical conditions were as follows: column temperature, 37 °C; mobile phase, 5 mM $H_2SO_4$; flow rate, 0.6 mL/min; detection, UV 210 nm. In order to determine the enantiomeric purity of LA in the post-fermentation broth, the following enzymatic tests were used from Nzytech: the D-/L-LA, UV method (Lisboa, Portugal). The optical purity of D-LA was defined as follows: D-LA/(L-LA + D-LA) × 100% [25].

## 3. Results

### 3.1. Effects of Various Nitrogen Sources on D-LA Biosynthesis

Table 1 shows the nitrogen content and the amount of substrate used for the fermentation medium for 19 selected inorganic and organic raw materials. The amount of substrate used in the production medium corresponds to the nitrogen content of 0.5 g/L.

The results of D-LA biosynthesis under stationary conditions on media containing an appropriate amount of organic or inorganic substrates with the nitrogen content of 0.5 g/L are presented in Table 2. Both organic and inorganic sources of nitrogen can be used as substitutes for yeast extract in the D-LA biosynthesis process.

Among the tested substrates, seeds of legumes such as pea, alfalfa, lupine and soybean were selected as optimal. The amount of D-LA obtained in the medium based on pea seeds hydrolysate after 120 h of biosynthesis was 81.5 g/L, while the amount of D-LA obtained in the medium containing yeast extract after 96 h of biosynthesis was 83.6 g/L. The amounts of D-LA obtained in the media based on hydrolysates of ground alfalfa, lupine and soybean seeds after 120 h were 61.0 g/L, 74.4 g/L and 63.6 g/L, respectively. The use of other organic raw materials as a source of nitrogen in the fermentation medium made it possible to obtain amounts of D-LA ranging from 17.9 to 51.3 g/L in the period from 96 to 168 h. The use of inorganic substrates in the form of ammonium salts, $(NH_4)_2SO_4$ and $NH_4NO_3$, caused low D-LA biosynthesis amounting to 33.0 and 8.9 g/L of D-LA, respectively. In all cultures, small amounts of by-product acetic acid were found, the concentration of which remained at the level of 0.5–3.0 g/L. It was further found that the enantiomeric purity of D-LA produced during the biosynthesis was close to 100%. Further studies using pea seeds were conducted to check whether a two-, three-, four- and five-fold increase in its content in the medium increased the performance of D-LA biosynthesis process. The studies also assessed whether the *Sp. laevolacticus* DSM 442 strain is able to use the nitrogen contained in pea seeds without prior enzymatic hydrolysis. The results of the abovementioned experiments are presented in Figure 1.

**Table 2.** D-LA Biosynthesis by *Sp. laevolacticus* DSM 442 strain under stationary conditions on media containing inorganic and organic substrates that are a source of nitrogen.

| Source of Nitrogen | Optimal Fermentation Time [h] | Unreacted Glucose Content [g/L] [a] | D-LA Concentration [g/L] [a] | D-LA Yield [%] [b] |
|---|---|---|---|---|
| $(NH_4)_2SO_4$ | 96 | 63.4, 0.26 | 33.0, 0.56 | 33 |
| $NH_4NO_3$ | 96 | 89.3, 0.59 | 8.9, 0.22 | 9 |
| Yeast extract | 96 | 12.8, 1.55 | 83.6, 2.65 | 84 |
| Urea | 120 | 79.4, 0.69 | 17.9, 1.29 | 18 |
| Meat extract | 96 | 62.8, 1.09 | 34.6, 2.09 | 35 |
| Peptone Bacteriological | 96 | 55.3, 1,45 | 36.0, 0.09 | 36 |
| Casein tryptone | 96 | 54.8, 1.22 | 42.9, 0.45 | 43 |
| Yeast cake | 168 | 45.2, 1.25 | 51.3, 0.78 | 51 |
| Corn steep liquor | 168 | 60.2, 1.65 | 36.9, 0.09 | 37 |
| Baker's yeast | 168 | 50.2, 1.59 | 46.2, 2.51 | 46 |
| Wheat middling | 144 | 52.8, 1.22 | 44.1, 1.65 | 44 |
| Hazelnuts | 120 | 60.7, 1.68 | 34.8, 1.89 | 35 |
| Pumpkin seeds | 120 | 60.5, 2.09 | 35.5, 0.99 | 36 |
| Flax seeds | 120 | 66.2, 1.98 | 30.1, 0.89 | 30 |
| Sunflower seeds | 120 | 59.4, 0.98 | 37.6, 0.78 | 38 |
| Alfalfa seeds | 120 | 36.2, 0.88 | 61.0, 1.89 | 61 |
| Lupine seeds | 120 | 21.20, 0.5 | 74.4, 0.89 | 74 |
| Pea seeds | 120 | 15.3, 0.23 | 81.5, 2.90 | 82 |
| Soybean seeds | 120 | 34.9, 0.89 | 63.6, 3.09 | 64 |

[a] $n = 3$, mean value, standard deviation, [b] calculated from the amount of glucose introduced into the reaction (summary equation: $C_6H_{12}O_6 \rightarrow 2 \times C_3H_6O_3$).

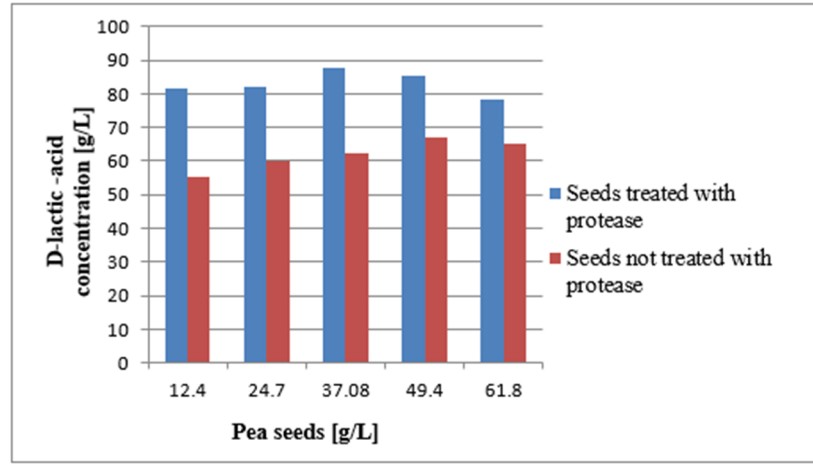

**Figure 1.** Influence of prior enzymatic hydrolysis on the production of D-LA by *Sporolactobacillus laevolacticus* DSM 442 strain; the bar graph was prepared on the basis of the mean values ($n = 3$).

The increase in the pea content in the medium from 12.4 g to 61.8 g/L did not result in a significant increase in the concentration of D-LA in the broth after 5 days of the fermentation process. For all pea concentrations tested, the content of D-LA ranged from 81.4 to 88.7 g/L. However, the enzymatic hydrolysis process itself had a significant influence on the increase in D-LA concentration. The prior treatment of protein material using protease resulted in an increase in D-LA concentration at the average level of 23.4 g/L for all tested pea concentrations. In order to further evaluate the process of D-LA biosynthesis by the *Sp. laevolacticus* DSM 442 strain, studies were carried out in batch cultures and in feed-batch cultures in bioreactors on fermentation media containing the optimum amount of nitrogen (0.5 g/L).

### 3.2. Batch Fermentation

The results of the D-LA biosynthesis process in batch cultures on media containing yeast extract and pea seeds as a nitrogen source are presented in Figure 2a,b.

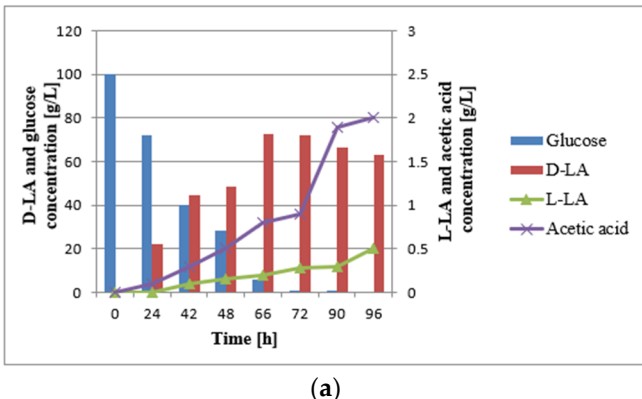
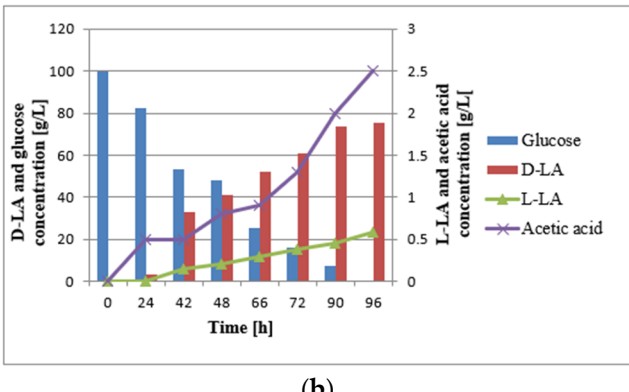

(**a**)
(**b**)

**Figure 2.** Batch fermentation by *Sp. laevolacticus* DSM 442 strain on the substrate containing yeast extract (**a**) and pea hydrolysate (**b**); the bar graphs were prepared on the basis of the mean values (*n* = 3).

As a result of a batch culture, which was conducted until the carbon source was completely depleted (after 72 h), 72.6 g/L of D-LA was obtained on the medium containing yeast extract (productivity 1.1 g/L/h). When the pea hydrolysate was used, after 96 h, 75.5 g/L of D-LA was obtained (productivity 0.79 g/L/h). When the yeast extract, as well as the hydrolyzed pea were used as the nitrogen source, the enantiomeric purity of the obtained D-LA was 99.7% and 99.4%, respectively. It was also found that the acetic acid concentration gradually increased in both cultures. When the medium component was yeast extract, the maximum level of acetic acid concentration was 2 g/L, and it was 2.5 g /L if the pea seeds were among the ingredients.

### 3.3. Fed-Batch Fermentation

The batch fermentation process with a single supplementary feeding was carried out in a 7.5 L bioreactor on the media containing yeast extract and pea hydrolysate and the results are shown in Figure 3a,b.

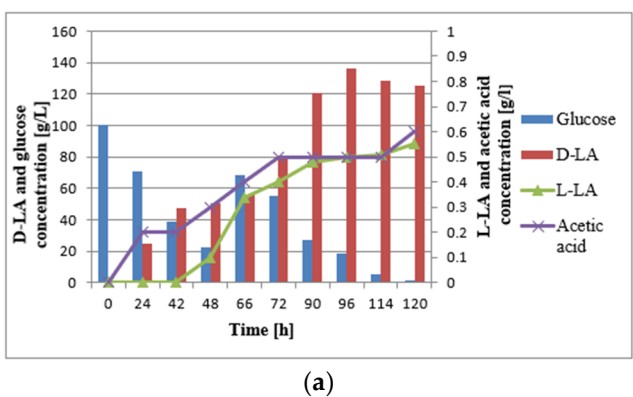
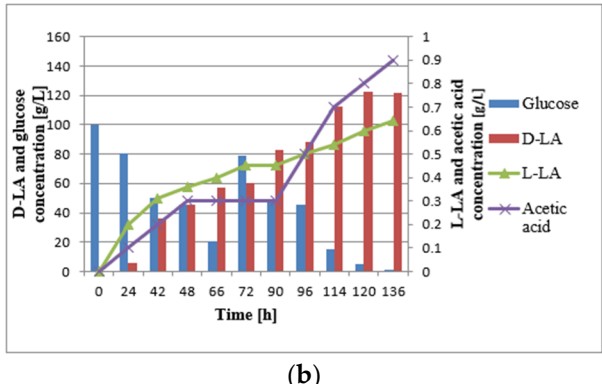

(**a**)
(**b**)

**Figure 3.** Fed-batch fermentation by *Sp. laevolacticus* DSM 442 strain on the substrate containing yeast extract (**a**) and pea hydrolysate (**b**); the bar graphs were prepared on the basis of the mean values (*n* = 3).

As shown in Figure 3, the process with the *v*/*v* yeast extract as the nitrogen source (Figure 3a) was completed after 96 h. The final D-LA concentration reached 136.5 g/L, and the glucose was completely exhausted. The yield, productivity and optical purity of D-LA were 97.5%, 1.42 g/L/h and 99.6%, respectively. When the pea hydrolysate was the nitrogen source (Figure 3b), the process was completed after 120 h. The final D-LA

concentration reached 122.6 g /L. The yield, productivity and optical purity were 87.6%, 1.021 g/L/h, 99.5%, respectively. It was also shown that the maximum concentration of acetic acid did not exceed 1 g/L in any of the culture used.

## 4. Discussion

Research shows that relatively low process yields and high production costs are the basic limitations of LA biosynthesis. Studies are being conducted worldwide to find the optimal production strain, as well as to reduce costs through the use of cheap substrates from agriculture or the food industry [26,37–39]. Lactic acid bacteria are characterized by complex development requirements, and in addition to carbohydrates, they need nitrogen sources in the form of amino acids, nucleic acids, as well as a complex of B vitamins and minerals for proper growth and development. According to some authors, these factors significantly affect the production of LA [40]. They also stimulate the growth of bacterial cells, their division, and influence their higher density in solutions [41].

The most commonly used nitrogen source for the biosynthesis of LA is yeast extract. However, this is a relatively expensive substrate, and its use as a component of the fermentation medium significantly reduces the profitability of industrial scale LA production. In our study, in the search for an alternative source of nitrogen for D-LA biosynthesis, we identified 19 organic and inorganic substrates, including food products (flax seeds, pumpkin seeds, hazelnuts), medium components (yeast extract, urea, meat extract, peptone bacteriological, casein tryptone), waste products from industry (corn steep liquor, slurry yeast, wheat middlings), as well as seeds of leguminous plants (peas, lupine, alfalfa and soy), which are very common in Poland, in the amount equivalent to the amount of nitrogen in 5 g of yeast extract (0.5 g N).

Our results confirm that yeast extract is the optimal substrate for D-LA biosynthesis by the *Sp. laevolacticus* strain, and the amount of D-LA obtained by biosynthesis under stationary conditions after 96 h was 83.6 g/L. Hydrolyzed pea seeds turned out to be an equally promising substrate for D-LA biosynthesis by the test strain. The amount of D-LA obtained in post-fermentation broth based on pea seed hydrolysate during biosynthesis under stationary conditions after 120 h was 81.4 g/L.

The possibility of using hydrolyzed pea flour to replace yeast extract and peptone in MRS medium was reported by Altaf et al. [24]. Their research showed that although *Lactobacillus amylovorus* GV 6 strain can use peas as a nitrogen source in the L-LA biosynthesis process, baker's yeast and red lentil were a much better alternative.

According to some authors, increasing the nitrogen substrate content may significantly increase D-LA biosynthesis [22,25]. Our research has shown that increasing the amount of nitrogen in the medium with pea hydrolysate (by two, three, four and five times) did not significantly increase D-LA biosynthesis, which can considerably reduce the cost of the process. We also checked if it was possible to use protein substrate directly, without its prior enzymatic hydrolysis, which would additionally contribute to the reduction of process costs. Our research has shown that although *Sp. laevolacticus* DSM 442 strain produces its own proteolytic enzyme, its prior hydrolysis with Neutrase 0.8 L in the amount of 0.5 g/L positively affects the yield of the biosynthesis process. Our results are consistent with those of Hsieh et al. [42], who found that prior hydrolysis of soybean protein significantly influenced the effectiveness of L-lactate biosynthesis by *Lactobacillus amylovorus* strain. According to the available literature, there are numerous microorganisms capable of producing L-LA, while D-LA is produced only by a few strains with relatively low yields [2,43–46]. However, according to our knowledge, there is little research on the production of D-LA by the *Sp. laevolacticus* DSM 442 strain [1,25,47]. Its nutritional requirements seem to be lower compared to other commonly used LA bacteria [48], which makes the *Sp. laevolacticus* strain a promising candidate for the production of D-lactate from raw agricultural materials. A high concentration of D-LA was obtained in batch fermentation (75.5 g/L) and fed-batch fermentation (122.6 g/L) with the use of pea seeds

as the sole source of nitrogen, indicating the huge potential of *Sp. laevolacticus* DSM 442 and the possibility of using it for D-LA production.

## 5. Conclusions

Our research has shown that the *Sp. laevolacticus* DSM 442 strain is able to produce D-LA on a substrate based on hydrolyzed pea seeds, which could be a promising nitrogen source for industrial scale production of D-LA. This eliminates the need for using expensive yeast extract as a source of nitrogen. The yield of D-LA using pea seed hydrolysate is comparable to the yield of D-LA using yeast extract. Moreover, the use of pea seed hydrolysate does not require the use of additional nitrogen sources. Owing to the high yield and optical purity, the obtained D-LA acid has significant potential for industrial applications such as the production of polymers or plant protection agents.

**Author Contributions:** Conceptualization, A.K.M. and A.B.; methodology, A.K.M., S.G., B.M.; formal analysis, A.K.M.; investigation, A.K.M., S.G., A.B.; writing—original draft preparation, A.K.M.; writing—review and editing, A.K.M., J.Z.; visualization, A.K.M. and J.Z.; supervision, A.B., J.Z. All authors have read and agreed to the published version of the manuscript.

**Funding:** The research was carried out under the Strategic Programme for Scientific Research and Development Works "Environment agriculture and forestry"—BIOSTRATEG/Competition II, No. 2/298338/2/NCBR financed by the National Centre for Research and Development.

**Institutional Review Board Statement:** Not applicable.

**Informed Consent Statement:** Not applicable.

**Data Availability Statement:** The data presented in this study are available on request from the corresponding author (A.M.).

**Conflicts of Interest:** The authors declare no conflict of interest.

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
