# Peer review of "Influence of Nitrogen Sources on D-Lactic Acid Biosynthesis by Sporolactobacillus laevolacticus DSM 442 Strain"

_fermentation, doi:10.3390/fermentation7020078_

Round 1
Reviewer 1 Report
1. In lines 101-104, the volumes of “Erlenmeyer flask and bioreactor” used are not
described.
2. In lines 146 and 157, “0.8L vvm,” and “0.8 vvm,” are different, respectively.
3. In sections 2-8 (Batch fermentation) and 2-9 (Fed-batch fermentation), what kinds data
are collected and then presented in Figs. 2 and 3 are not understood. What is the data
showed on the right vertical axis, i.e. on the opposite site of “Concentration”. In
addition, the unit of left vertical axis Fig. 2(b) should be “g/L”.
4. In section 2-9 (Fed-batch fermentation), the time for addition of high glucose medium is
not described, which is important for fed-batch fermentation.
5. In line 165, “--- at 8000 rpm ---” should exhibited as “AAA x g”.
6. In lines 181-182. “a) calculated value, b) nitrogen content provided by a supplier. c)
nitrogen determination by Kjeldahl method. Each value is mean ± standard deviation
(n = 3),”, the “period (.)” after “supplier” mat be right. Also, “a)” and “b)” could not
have “n = 3”. In addition, what is the “order” of these N-source of “Types of raw
material”?
6. The complicated changes of residual glucose and produced lactic acid, in batch
fermentation and fed-batch fermentation, are not well stated in lines 223-231 and lines
224-245.
7. In line 274, “--- reported by Altaf et al. [40].” is not found in “References”.
8. Many part of “References” the “Genus” and “Species” names of organisms and
microorganisms are not be italicized. For example, lines 322 (Lactobacillus
delbrueckii), 378 (Sporolactobacillus inulinus), and 428 (Lactobacillus bulgaricus).
Author Response
Dear Editor, The authors would like to thank the reviewers for a careful examination of our manuscript and for valuable remarks which will undoubtedly enhance the value of our work. The answers to the reviewers’ comments are presented in Table below. The corrections in the manuscript are marked in yellow. Best regards, Alicja Katarzyna Michalczyk, Jerzy Zakrzewski, Apr. 25, 2021 R1 The abbreviation of lactic acid (LA) as defined in line 1 could be used throughout the text Done R1 In introduction section I suggest including the motivation for the strain choice. In details a little overview about the use of Sporolactobacillus laevolacticus DSM 442 for LA production should be included Done The choice of Sp. laevolacticus DSM 442 as a strain enabling the production of D-LA was dictated by its low nutritional requirements during the fermentation process. More detailed information – see Discussion section. R1 Line 73-79 should be included in the conclusion section and not in the introduction one Included in the conclusion; lines 468-474 R1 Please correct the English structure in line 113 Done R1 Please add a reference for media composition Done The references for media composition added: ref. 1, ref. 36. R1 Please explain or add a reference regarding the use of air flow in batch and fed-batch fermentation The optimal values for air flow in batch and fed-batch fermentation are presented in lines 223 and 241 R1 In my opinion pH control during long fermentation processes is important. Did you check this parameter? If not, why? Done. The pH value was checked throughout the whole process of batch and fed-batch fermentation, and remained at the level of approx. 5.5-5.7 (adjusted with 60% CaCO3). Information: lines 225-228. R1 Please add details about "possible by-products" in line 148 Apart from lactic acid, small amounts of acetic acid may be formed during lactic acid fermentation. Information about acetic acid as a by-product: lines 228, 251 R1 Please add details for sampling time in bed-batch fermentation Done R1 Check the values of total nitrogen content and amount of substrate used in table 1. I think that there are some errors (e.g. urea and alfa alfa values there is no correspondence) Done R1 Please add statistical significance and standard deviation in figures and statistical significance in tables Table 2 and figures were edited using the mean values (n=3). R1 Please add second vertical axis name in figure 2 and 3 Done R1 Please delete minus before value 2.5 in line 231 Done R2 The title can be simplified: Influence of nitrogen sources on D. lactic… Done Influence of nitrogen sources on D-lactic acid biosynthesis by Sp. laevolacticus DSM 442 strain R2 Line 40: polylactide: is it made only with D-lactic acid, can you clarify? PLA are synthesized from both D- as well as L-lactic acid; explained in line 62, R2 Line 48: yeast extract, introduced (YE) Done YE was changed to yeast extract R2 Line 66: LA: could you clarify, L or D lactic acid? Both stereoisomers of LA; explained in line 83 R2 Line 76: Simplify: The quantity of D-LA measured on the fermentation medium…. Done R2 Line 96: Sporolactobacillus laevolacticus is a producer of homofermentative D-lactic acid, why do you also measure L-lactic acid, albeit in minor amounts. Processes catalyzed by microorganisms are stereoselective, but generally not in 100%. The production of D-LA by Sp. laevolacticus is stereoselective. D-LA is a basic product, although small amounts of L-LA are produced as well. R2 Line 113: Simplify: Fermentation media containing an organic and inorganic nitrogen source in the amount specified in Table 1 as well as …………… ..FeSO4x7H2O were prepared. Done R2 Line 121-122: deletion obtained in both sentences. In fact obtained could be supressed in most sentences (line 135, 267, 268, 271, 285…) Done R2 Paragraph 2.6 and throughout the text and in the tables 1, 2, all results should be given with a decimal point, e.g. 12.36 g / l should be 12.4 g / l Done R2 Line 147 : rephrase : Two samples were collected …..and possible by-products at different times. Before it was: Every few hours two samples were collected to determine the content of LA (D-LA, L-LA), unreacted glucose and possible by-products. Now is: Two samples were collected to determine the content of LA (D-LA, L-LA), unreacted glucose and possible by-products at different times. R2 Line 164 : Samples were treated with one volume (suppress taken periodically) Done Samples were treated with one volume of 0.5 M H2SO4 and centrifuged at 8000 rpm (Universal 320R, Hettich, Germany) for 30 min. R2 Results: Subtitles should be added as in materials and methods. Done R2 Figures 2: 2 a and 2 b must have the same scales (right 120 Ok and left 2.5 or 3 for both). The value must be added on the left scale (g/l?). It should be specified that for Glucose in D-LA, the reference scale is on the left part, and for L-LA and acetic on the right part Done R2 Same comment for the Figure 3 Done R2 Line 267 : Our results confirm … Done R2 Line 274 : the name of the bacteria should be written Lactobacillus amylovorus in italic and then line 288 L. amylovorus Done R2 Line 281 : We also checked if it was possible to use :… Done R2 Line 285 : Our results are consistent with the ones of Done

Reviewer 2 Report
This work is well organized and the topic is interesting. The organization of different steps is clear and the presentation of results is easy to read.
Nevertheless, I think that the manuscript should be improved in some details:
- The abbreviation of lactic acid (LA) as defined in line 1 could be used throughout the text
- In introduction section I suggest including the motivation for the strain choice. In details a little overview about the use of Sporolactobacillus laevolacticus DSM 442 for LA production should be included
- Line 73-79 should be included in the conclusion section and not in the introduction one
- Please correct the English structure in line 113
- Please add a reference for media composition
- Please explain or add a reference regarding the use of air flow in batch and fed-batch fermentation
- In my opinion pH control during long fermentation processes is important. Did you check this parameter? If not, why?
- Please add details about "possible by-products" in line 148
- Please add details for sampling time in bed-batch fermentation
- Check the values of total nitrogen content and amount of substrate used in table 1. I think that there are some errors (e.g. urea and alfa alfa values there is no correspondence)
- Please add statistical significance and standard deviation in figures and statistical significance in tables
- Please add second vertical axis name in figure 2 and 3
- Please delete minus before value 2.5 in line 231
Author Response

(The authors gave the same response as above.)

Reviewer 3 Report
The authors studied the influence of different sources of nitrogen on the production of D. lactic acid by Sporolactobacillus laevolacticus. They have shown that a high level can be produced using pea hydrolyzate. The manuscript is easy to read, well structured and the methodology appropriate. However, the following remarks can be addressed:
The title can be simplified: Influence of nitrogen sources on D. lactic…
Line 40: polylactide: is it made only with D-lactic acid, can you clarify?
Line 48: yeast extract, introduced (YE)
Line 66: LA: could you clarify, L or D lactic acid?
Line 76: Simplify: The quantity of D-LA measured on the fermentation medium….
Line 96: Sporolactobacillus laevolacticus is a producer of homofermentative D-lactic acid, why do you also measure L-lactic acid, albeit in minor amounts.
Line 113: Simplify: Fermentation media containing an organic and inorganic nitrogen source in the amount specified in Table 1 as well as …………… ..FeSO4x7H2O were prepared.
Line 121-122: deletion obtained in both sentences. In fact obtained could be supressed in most sentences (line 135, 267, 268, 271, 285…)
Paragraph 2.6 and throughout the text and in the tables 1, 2, all results should be given with a decimal point, e.g. 12.36 g / l should be 12.4 g / l
Line 147 : rephrase : Two samples were collected …..and possible by-products at different times.
Line 164 : Samples were treated with one volume (suppress taken periodically)
Results: Subtitles should be added as in materials and methods.
Figures 2: 2 a and 2 b must have the same scales (right 120 Ok and left 2.5 or 3 for both). The value must be added on the left scale (g/l?). It should be specified that for Glucose in D-LA, the reference scale is on the left part, and for L-LA and acetic on the right part
Same comment for the Figure 3
Line 267 : Our results confirm …
Line 274 : the name of the bacteria should be written Lactobacillus amylovorus in italic and then line 288 L. amylovorus
Line 281 : We also checked if itw as possible to use :…
Line 285 : Our results are consistent with the ones of
Author Response

(The authors gave the same response as above.)

Round 2
Reviewer 1 Report
Reviewer’s Comments: (R3)
1. R2. Line 40: polylactide: is it made only with D-lactic acid, can you clarify?
Authors’ response. PLA are synthesized from both D- as well as L-lactic acid; explained
in line 62,
R3. There is no explanation observed in line 62.
Line 62 of 1200384.v2, “In order to reduce the costs of the process, there were attempts
made to use cheap”.
2. In Authors’ response to the comments of R3.
- In lines 101-104, the volumes of “Erlenmeyer flask and bioreactor” used are not
described.
Authors’ response. The volumes of the Erlenmeyer flask and bioreactors are given in
section 2.6. (thired sentence), 2.7 (thired sentence) and 2.8 ( first sentence).
R3. This response should be corrected to:
“section 2.6. (the 4th sentence), 2.7 (the 7th sentence)”
3. In Authors’ response to the comments of R3.
6a. --- In addition, what is the “order” of these N-source of “Types of raw material”?
Authors’ response. The changes were made in response to Reviewer ‘s 1 and Reviewer’s
2 comments.
R3. There is no related comments found in Reviewer ‘s 1 and Reviewer’s 2 comments.
What is he meaning of the “order” of these N-source of “Types of raw material” in
Table 1.
4. In Authors’ response to the comments of R3.
- Many part of “References” the “Genus” and “Species” names of organisms and
microorganisms are not be italicized. ---
Authors’ response. The required changes were made
R3. Most problems are fixed, but in line 351, “Geobacillus stearothermophilus” and
line 391, “Lactobacillus rhamnosus” still need to be fixed.
5. R3. In lines 351-352, the Journals’ names “Journal of Chemical Technology &
Biotechnology” and line 392, “Enzyme Microb Technol,” are differ from the format of
the rest Journals’ name in “References”, which are need to be corrected.
6. Could Authors’ explain why as “fresh medium containing 260 g/L of glucose” (line 166),
but in Fig. 3a and 3b, the glucose concentration was 100 g/L?
7. Could Authors’ explain why after addition of glucose, the glucose concentration in Fig. 3a
and 3b are different, which are approximately 70 and 80, respectively?
Author Response
Dear Editor, Below, please find the answers and comments to the Reviewer’s 3 remarks. Old corrections and comments – still in yellow.New corrections and comments – in blue. Reviewer 3 Answers 1. R2. Line 40: polylactide: is it made only with D-lactic acid, can you clarify?Authors’ response. PLA are synthesized from both D- as well as L-lactic acid; explained in line 62, R3. There is no explanation observed in line 62. Line 62 of 1200384.v2, “In order to reduce the costs of the process, there were attempts made to use cheap”. The text where the synthesis of polylactides from both lactic isomers is placed directly after references [1-10] in the Introduction part and is marked in blue. 2. In Authors’ response to the comments of R3. In lines 101-104, the volumes of “Erlenmeyer flask and bioreactor” used are not described. Authors’ response. The volumes of the Erlenmeyer flask and bioreactors are given in section 2.6. (the third sentence), 2.7 (the third sentence) and 2.8 (the first sentence). R3. This response should be corrected to: “section 2.6. (the 4th sentence), 2.7 (the 7th sentence)” The data concerning the volumes of the Erlenmeyer flasks and reactors were thoroughly checked. They are presented in the appropriate places of the sections 2.6, 2.7, and 2.8. 3. In Authors’ response to the comments of R3. 6a. --- In addition, what is the “order” of these N-source of “Types of raw material”? Authors’ response. The changes were made in response to Reviewer ‘s 1 and Reviewer’s 2 comments. R3. There is no related comments found in Reviewer ‘s 1 and Reviewer’s 2 comments. What is he meaning of the “order” of these N-source of “Types of raw material” in Table 1. + Preceding comment with the preceding number 6a In lines 181-182. “a) calculated value, b) nitrogen content provided by a supplier. c) nitrogen determination by Kjeldahl method. Each value is mean ± standard deviation (n = 3),”, the “period (.)” after “supplier” mat be right. Also, “a)” and “b)” could not have “n = 3”. In addition, what is the “order” of these N-source of “Types of raw material”? - The number of repetitions for the determination of a nitrogen by Kjehddal method was equal to n = 3. It is written in a footnote c) under Table 1; marked. in blue. - The order of nitrogen sources in Table 1 is given on the basis of the method of determining their nitrogen content: 4. In Authors’ response to the comments of R3. Many part of “References” the “Genus” and “Species” names of organisms and microorganisms are not be italicized. ---Authors’ response. The required changes were made. R3. Most problems are fixed, but in line 351, “Geobacillus stearothermophilus” and line 391, “Lactobacillus rhamnosus” still need to be fixed. “Geobacillus stearothermophilus” and “Lactobacillus rhamnosus” were italicized and marked in blue in the manuscript. The manuscript was thoroughly checked, all strain names are italicized. 5. R3. In lines 351-352, the Journals’ names “Journal of Chemical Technology & Biotechnology” and line 392, “Enzyme Microb Technol,” are differ from the format of the rest Journals’ name in “References”, which are need to be corrected. All references were re-written as required according to the “Instruction to the Authors”. The names of journals were italicized and correctly abbreviated. The changes are marked in blue. 6. Could Authors’ explain why as “fresh medium containing 260 g/L of glucose” (line 166),but in Fig. 3a and 3b, the glucose concentration was 100 g/L? + 7. Could Authors’ explain why after addition of glucose, the glucose concentration in Fig. 3a and 3b are different, which are approximately 70 and 80, respectively? In fed batch fermentation process, the initial (t=0 h) amount of glucose was 300 g. When the glucose concentration decreased below 20g/L, fresh medium (1 L) containing 260 g of glucose was added once to increase the glucose concentration in the medium at the level of 70-80 g/L. Above explanation was clearly presented in the Experimental Part:, section 2.8. The sections 2.7 and 2.8 were re-written and marked in blue. Sincerely yours, Alicja Katarzyna Michalczyk, Jerzy Zakrzewski, May 10. 2021
Round 3
Reviewer 1 Report
No further comments.